# Ultraviolet Photodetecting and Plasmon-to-Electric Conversion of Controlled Inkjet-Printing Thin-Film Transistors

**DOI:** 10.3390/nano10030458

**Published:** 2020-03-04

**Authors:** Cheng-Jyun Wang, Hsin-Chiang You, Jen-Hung Ou, Yun-Yi Chu, Fu-Hsiang Ko

**Affiliations:** 1Department of Materials Science and Engineering, National Chiao Tung University, 1001 University Road, Hsinchu City 30010, Taiwanharveyou@hotmail.com.tw (J.-H.O.); taipei.wayne@gmail.com (Y.-Y.C.); 2Department of Electronic Engineering, National Chin-Yi University of Technology No. 57, Sec. 2, Zhongshan Road, Taiping District, Taichung City 41170, Taiwan; hcyou@ncut.edu.tw

**Keywords:** ink-jet printing, zinc-oxide-based thin-film transistors, visible light photodetection, oxygen plasma treatment, plasmon energy detection

## Abstract

Direct ink-jet printing of a zinc-oxide-based thin-film transistor (ZnO-based TFT) with a three-dimensional (3-D) channel structure was demonstrated for ultraviolet light (UV) and visible light photodetection. Here, we demonstrated the channel structures by which temperature-induced Marangoni flow can be used to narrow the channel width from 318.9 ± 44.1 μm to 180.1 ± 13.9 μm via a temperature gradient. Furthermore, a simple and efficient oxygen plasma treatment was used to enhance the electrical characteristics of switching *I*_ON_/*I*_OFF_ ratio of approximately 10^5^. Therefore, the stable and excellent gate bias-controlled photo-transistors were fabricated and characterized in detail for ultraviolet (UV) and visible light sensing. The photodetector exhibited a superior photoresponse with a significant increase of more than 2 orders of magnitude larger drain current generated upon UV illumination. The results could be useful for the development of UV photodetectors by the direct-patterning ink-jet printing technique. Additionally, we also have successfully demonstrated that a metal-semiconductor junction structure that enables plasmon energy detection by using the plasmonic effects is an efficient conversion of plasmon energy to an electrical signal. The device showed a significant variations negative shift of threshold voltage under different light power density with exposure of visible light. With the ZnO-based TFTs, only ultraviolet light detection extends to the visible light wavelength.

## 1. Introduction

Over the past decade, field-effect transistors (FETs) and thin-film transistors (TFTs) based on organic and inorganic materials have been widely developed for advanced applications, such as flat-panel displays for realizing high-resolution displays. The advantages of TFTs based on metal-oxide-based materials include high electron mobility, thermal stability and stable electrical characteristics [1,2,3,4,5,6]. Recently, solution processing methods under low temperatures of 200–300 °C have attracted much interest for fabricating metal-oxide-based TFTs. Compared to the methods of traditional vacuum system deposition, solution-processing methods have a nature of simple processing, low temperature and low manufacturing cost [7,8,9]. Generally, the methods of low-temperature solution processing of devices include screen printing, spin coating, gravure printing and ink-jet printing [10,11,12,13]. Previous studies have reported direct printing as the most efficient method for the deposition of active materials onto appropriate positions without the necessary etching process and mask alignment, which reduces the involvement of toxic chemistry and energy requirements for the fabrication of thin-film transistor (TFT) arrays used in large-area electronic devices. However, these studies usually only demonstrated results of a single TFT device, indicating that significant difficulties may still exist for the transition from single devices to array devices. To form consistent channels for the study of array devices, ink-jet printing (IJP) was selected for the deposition of functional materials which is a technique that can achieve patterned microscale film morphologies without additional complication and expensive photolithography for the fabrication of an advanced-structure of electronic devices [14,15,16]. Moreover, capillary flow is a physical phenomenon that is frequently observed in printing films. It is highly undesirable in most cases because the flow phenomenon causes density fluctuations and makes it difficult for the realization of uniform film and clean patterns. A well-known result of capillary flow is the “coffee-stain” distribution or “coffee-ring effect”; this flow phenomenon also has a negative effect on the morphology and electrical characteristics of patterned devices. Therefore, processing techniques were developed to eliminate this effect and the coffee-ring effect can be effectively controlled [17,18,19]. However, recently a few studies have attempted to exploit the morphology of such effects and create patterned microstructures such as conductive patterned silver nanoparticles, a TFT based on a one-step fabrication route, and short channels of TFTs defined by coffee-rings [20,21,22].

Many studies have made several attempts to develop a metal-oxide semiconducting-based photodetecting. Metal-oxide semiconducting materials-based TFTs are considered to be the most appropriate photodetecting devices because of their characteristics of being transparent, low-cost and fast photoresponse in the ultraviolet wavelength region [23,24,25,26,27,28]. However, the metal-oxide semiconducting materials based photodetector can convert only ultraviolet (UV) wavelength light to electrical signal. This is because the intrinsic wide band gap of an oxide semiconductor material causes limitations of the absorbed light to the high-energy UV region. One of the key functions for using a plasmonic effect is an efficient conversion of plasmon energy to an electrical signal. Metallic nanoparticles exhibit unique surface plasmonic properties due to the abundant conduction electrons on their particle surface. A metal-semiconductor junction structure enables plasmonic energy detection as an electrical signal or current. Photos are absorbed in the metal nanostructure and generate hot electrons and plasmon energy resulting in photocurrent or energy band modification in semiconducting devices [29,30,31]. This phenomenon has significance in the development of photodetectors, which can generate photoresponse in devices.

In this work, we demonstrated a low-temperature processing method and facile strategy to create three-dimensional (3-D) channel structures of zinc-oxide (ZnO)-based thin-film transistors (TFTs) in various patterns by utilizing a solution process based on the coffee-ring effect derivatives with an ink-jet printing technique. These fabricated TFTs demonstrated applicability as sensitive ultraviolet (UV) photodetectors. This experiment showed the following four points of interest: (i) the effects of the channel arrays on TFT electrical characteristics by varied channel pattern widths miniaturized by ink-jet printing onto temperature-controlled substrates. (ii) The electrical performance improvements through efficient and facile oxygen plasma treatment. (iii) Realization of a low-cost, low-temperature, adjustable IJP techniques for ZnO films as excellent visible-blind UV sensitizers in optoelectronics, successfully demonstrating gate bias controlled photoinduced current amplifications. (iv) In order to increase the photoresponse under visible light, gold nanoparticles (AuNPs) were used for efficient conversion of plasmon energy in this visible wavelength photodetecting. The device showed significant variations in negative shift of threshold voltage under different light power density with exposure of 520 nm visible light. In addition, the printing method and enhanced electrical technique employed for the fabrication of the three-dimensional channel structure of the ZnO-based thin film transistor UV photodetector is valuable and promising for non-lithographical processing of optoelectronic devices.

## 2. Materials and Methods

### 2.1. The Semiconducting Material of Zinc-Oxide Solution

To prepare the semiconducting channel solution, the transparent conducting oxide solution of zinc oxide (ZnO) was prepared by dissolving zinc acetate dehydrate [Zn(CH_3_COO)_2_ 2H_2_O] in ethanol (CH_3_CH_2_OH, absolute 99.8%) as precursors with the concentration of 0.05 M (All chemicals are obtained from Union Chemical, Taipei, Taiwan). Then, the solution was gently stirred for 1 h at 60 °C. Finally, a high-transmittance solution of ZnO was obtained and is shown in the inset of Figure 1a.

### 2.2. The Device Fabrication

The ZnO-based TFTs of the 3-D channel structure were fabricated using a heavily p-doped silicon substrate as the bottom gate, and a 100-nm-thick silicon nitride (Si_3_N_4_) layer was deposited on a silicon substrate by plasma-enhanced chemical vapor deposition (PECVD) system at 300 °C processing. The ZnO precursor solution was inkjet-printed on the dielectric layer by using a Dimatix materials printer (DMP-2800 Series) as an active material of the n-type transistor, and at the same time, ink-jetting ZnO solution and the substrate holder were adjusted to room temperature (RT), 40 and 60 °C, respectively. Subsequently, the ZnO thin film was processed by air annealing at 300 °C for 1 h. Finally, source/drain and bottom gate electrodes regions were defined by metal masks followed by thermal coater deposition of a 300-nm-thick film of aluminum, as shown in Figure 1b. The electrodes length and width of the thin film transistor defined by the metal mask were 70 μm and 2000 μm, respectively (the devices manufacturing from Taiwan Semiconductor Research Institute, Taipei, Taiwan).

## 3. Results and Discussions

### 3.1. Coffee-Ring Structure of 3-D Narrow Channel Formation

The ink-jet printing technique was employed for large-area batch processing and used for the direct design of various patterns for the fabrication of the semiconducting channel layer. Herein, we demonstrated channel structures formed through the coffee-ring effect, and that temperature-induced Marangoni flow can be used to narrow the channel width via a temperature gradient, Figure 2a shows the optical microscope (OM) images at an optical magnification of 4 times and 20 times of the ink-jet-printed ZnO-based TFT fabricated on the silicon nitride (Si_3_N_4_) dielectric layer. The patterned ZnO solution exhibited an excellent composition of continuous thin-film channel width patterns between the source and drain electrodes. Additionally, the channel widths were both obviously narrower than the electrodes, indicating that the IJP technique could provide a route to directly pattern the semiconductor transistor channel widths and fabricate the microscale transistor circuits without any complicated processing of the photolithography. Moreover, the evaporation behavior during the ZnO solution drying process played a vital role in controlling the film morphology and semiconducting channel width throughout the ink-jet-printed films. We evaporated the ZnO solution inkjet droplets deposited onto the Si_3_N_4_ dielectric layer at RT. The coffee-stain effect is a physical phenomenon that is frequently observed in printing films, such that when the ZnO solution in the drying process was transported out to form the deposition and created a relatively wide channel by the capillary flow effect, as shown in Figure 2b; the average value for the channel widths of the ZnO solution drying at room temperature were 318.9 ± 44.1 μm, which were the averaged values achieved from 10 devices. The contact angle measured using a deionized (DI) water droplet on the solid interfaces of the Si_3_N_4_ dielectric layer was 37.48° (as depicted in Figure 2c inset). Therefore, the control of the coffee-ring effect is necessary for the fabrication of high-quality and highly channeled pattern. One approach that can improve the homogeneity of the ink-jet-printed deposition involved the modulation of the substrate temperature to control the transport of the channel widths during the solvent evaporation.

Haena Kim et al. [32] and Xiaoying Shen et al. [19] reported that a rapid solution evaporation rate that exceeded the solvent transportation rate in a droplet (*υ*_transp_ < *υ*_evap_) reduced the capillary flow effect. On the other hand, the solution evaporation rate increased as the substrate temperature increased, and the capillary flow effect decreased, enabling the decrease in the channel width, as shown in Figure 2b. Figure 2c shows the results of distinct different depositing formation behaviors detected at the higher substrate temperature; the channel widths were narrowed to 255.2 ± 18.7 μm and 180.1 ± 13.9 μm at 40 °C and 60 °C, respectively (the 10 devices of channel width shown in Appendix A and Appendix A). A convective flow from the center toward the drying edge of the droplet creating a relatively wide channel was observed. By careful control of the coffee-effect phenomenon, devices can be improved through increasing the substrate temperature-induced Marangoni flow [33] and create narrow semiconducting channel widths. The channel width was narrowed from the maximum of 410 μm to the minimum of 160 μm in this experiment.

### 3.2. Electrical Characterization

Based on the channel morphology described above, we relate the changes of the transfer electrical characteristics of the ZnO-based TFT performances to the variation of the substrate temperature during the ink-jet printing procedures. The average value for the normalized threshold voltage (*V*_th_) of the TFTs prepared at room temperature, 40 and 60 °C were 4.01 ± 1.59, 3.87 ± 1.12 and 1.33 ± 1.06 V, respectively, as shown in Figure 3a, while Figure 3b shows that the average values of the field-effect electron mobility were 10.00 ± 3.18 × 10^−3^, 3.31 ± 2.04 × 10^−3^ and 37.20 ± 35.13 × 10^−3^ cm^2^/(V·s), respectively (the each devices of electrical characteristics shown in Appendix A). The electron mobility as determined according to the following drain current saturation equation:*I*_Drain,sat_ = (*W*/2 *L*) × *Ci* × µ_FE_ × (*V*_GS_ − *V*_th_)^2^(1)

Here, the *W* is the channel width, *L* is the channel length, *Ci* is the capacitance of gate insulator per unit area (~1.22 × 10^−8^ F/cm^2^, where in this experiment of silicon nitride thickness 100 nm), µ_FE_ is the electron mobility, *V*_GS_ is the voltage applied between gate and source electrodes, and *V*_th_ is the threshold voltage.

The difference in the electrical characteristics has arisen from the influence of the morphology of the semiconducting ZnO films on the electrical conductivity. The ZnO-based TFT without the substrate temperature heating preferred a resistive operation (linearity operation) rather than a TFT device operation, as shown in Figure 3c (black line). This indicated that the semiconducting channel was rough and did not provide an obvious electrons transport path, leading to the observed relatively worse electrical characteristics of unstable operation voltage to turn-on the TFT and lower electron mobility. However, the electrical performance was improved by increasing the substrate temperature during the ink-jet deposition of the ZnO solution. Appendix A depicts a ring-like channel wire widths depositing effect and the semiconductor within conduction carrier flowing behaviors. A more markedly capillary flow caused zinc oxide solution to be deposited on both sides of the periphery and formed a thin-film area higher than a central one; both sides (ring-like depositing) of resistance (R) values will be much less than the central region because the resistance cross-sectional area is much larger than central regions by the electrical resistance and leading semiconductor electron carriers flow from the peripheral ZnO-wire sides to dominate instead of conduction electrons through the center of the high-resistance region. Therefore, when the ZnO channel formation was narrowed, this created a smaller and stable threshold voltage to turn on the devices because of obvious and neat channel film of good contrast during the solution drying process by heating.

Although the fabrication of the ZnO-based TFTs by the ink-jet printing process was achieved, significant electrical challenges such as the significant *I*_ON_/*I*_OFF_ switching ratio and a stable threshold voltage (*V*_th_) remained in the fabricated electronics devices making them unsuitable for use as a photosensitive element. For the reasons stated above, a high-quality ZnO oxide channel film can be improved by solution deposition followed by a simple processing of O_2_ plasma treatment. Moreover, solution-based deposition followed by an O_2_ plasma treatment could be a desirable method of a low-temperature technique for manufacturing high-performance TFT device [34,35,36,37]. To comprehensively investigate the electrical characteristics of the ZnO-based TFTs after the O_2_ plasma treatment on the semiconducting ZnO channel film in TFTs, the typical output characteristics of the drain current-gate voltage (*I*_DS_–*V*_GS_) curves are shown in Figure 3c (green line). A significant improvement in electrical performance was obtained at the particular applied bias voltage of V_GS_ from −20 up to 20 V at the constant *V*_DS_ of 5 V after the O_2_ plasma treatment for one minute. We obtained a significant improvement in the switching *I*_ON_/*I*_OFF_ ratio of approximately 10^5^ and the electron mobility in the saturation region also increased to 263 × 10^−3^ cm^2^/(V·s), leading to the suitable demonstration of the ZnO-based TFTs in photosensitive devices.

Moreover, to determine the improvement of the electrical characteristics by the active oxygen ions in active channel layer induced by the oxygen plasma treatment, Figure 3d shows the X-ray photoelectron spectroscopy (XPS) spectra of the ZnO film before and after the O_2_ plasma treatment. Of most interest is that the O 1*s* peak at the surface that was consistently fitted by a Lorentzian–Gaussian function, and the components of O_I_, O_II_, and O_III_ were centered at ~530.6, ~531.9, and ~532.4 eV, respectively. The component O_III_ at the high binding energy located at 532.4 eV is mainly due to the presence of loosely bound oxygen on the surface of the ZnO film, such as –CO_3_, absorbed H_2_O, or absorbed O_2_ [38]. Obviously, the intensity of the O_III_ clearly increased which implied the O_2_ plasma treatment caused an accumulation of absorbed O_2_ on the ZnO surface after plasma treatment for 1 min. The intensity of O_III_/O_total_ was about 14.64% (O_I_, O_II_, O_III_ of ratio: 65.58%, 19.78%, 14.64%) for the ZnO film and increasing to 23% (O_I_, O_II_, O_III_ of ratio: 62.04%, 14.96%, 23%) for the ZnO film with oxygen plasma treatment. This observation confirmed that a clear increased concentration of the oxygen chemical state and the increased oxidation on the surface enhanced the electrical performance of the semiconducting channel layer. Further, negligible components change was observed with the O_2_ plasma treatment (Appendix A) on the binding energies of the Zn 2p_3/2_ and Zn 2p_1/2_ components were centered at ~1021.4 and ~1044.5 eV, respectively. This is because the oxygen O_2_ treatment primarily improves the surface quality of ZnO film, but has no significant effect on their microstructure and crystallinity [39].

### 3.3. ZnO-based Thin-Film Transistor (TFT) of Ultraviolet (UV) and Visible Light Photodetector Performance Characterization

The material of ZnO is an ideal ultraviolet sensitizer and broad-band transparent conductor for visible-blind ultraviolet (UV) detection because ZnO is a direct band gap semiconducting material with a band gap of 3.37 eV [40,41]. For photodetecting devices, the device stability and endurance of electrical characteristics were the most important consideration. Figure 4a,b showed the endurance of the ZnO-based TFT fabricated by direct ink-jet printing after oxygen plasma treatment as examined using the I_DS_–V_GS_ curves. The device showed excellent stable I_ON_/I_OFF_ switching electrical characteristics after 5 operations and the square root of the curves also demonstrated a stable electron mobility and threshold voltage (*V*_th_). For both electrical properties, *I*_DS_ is changed marginally after 5 operations. This indicated that this device has stable and excellent electrical properties for UV photodetection. Moreover, the ZnO material is a broad-band transparent conductor for visible-blind UV detecting. As shown in Appendix A of the transfer electrical characteristics of devices under illumination at different visible wavelengths of 500, 600 and 700 nm, the device electrical properties changed marginally under illumination. The inconspicuous response to the visible light in the region ranging from near 400~700 nm is mainly ascribed to the light absorption related to oxygen vacancies located at the subgap energy level near the valence band maximum of ZnO.

Figure 4c–f shows the UV photoinduced variations in the transfer curves of the ZnO-based TFTs electrical characteristics. When the ZnO absorbed light with energies higher than the band gap of 3.37 eV (UV wavelength), the device immediately excited the electron-hole pairs, generating a photocurrent in the photoactive ZnO channel layer. As shown in Figure 4c, significant threshold voltage shifts and off-current increases of more than 2 orders of magnitude were clearly observed immediately (red line). Moreover, the square root of the *I*_DS_–*V*_GS_ curves indicated the higher conductivity (higher slope performance, red line) under exposure to UV illumination, as shown in Figure 4d. This demonstrated the ability of the digital signal to readily switch between the “on” and “off” states synchronized with the UV light exposure. Figure 4d shows the transfer square root of the electrical characteristics; the *I*_DS_ is increased evidently under UV illumination, which is attributed to the photocarriers generated by the photoinduced electrons in the ZnO valence band moving to the conduction band due to the absorbed UV energy rather than to the electric field effect. In addition, the current versus voltage (*I*–*V*) curves of the semiconducting and photo-sensitive layer based on the ZnO layer under UV light illumination are shown Appendix A. Firstly, all the *I*–*V* curves are exhibited linear characteristics, it is indicating good ohmic contact formed between the ZnO channel layer and the aluminum electrodes. Secondly, it is found that the obvious photogenerated current increased from 0.71 to 10.30 μA at an applied bias voltage from −10 up to 10 V. Clearly, the ZnO channel layer not only provides excellent TFT and *I*–*V* operation characteristics but also provides a good photo-sensitive layer and transfer significant photo-generated current under the UV light illumination.

An intense photoresponse for the photoactive ZnO channel film was observed for illumination of 3.37 eV (UV light, <400 nm). In addition, the electrical properties were recovered to the initial state after the UV light was turned off, indicating that the photogenerated free carriers which were vigorously activated by light exposure were not trapped in the dielectric layer and caused an obvious threshold voltage shift, resulting in excellent electrical recovery properties. The typical drain current-drain voltage (*I*_DS_–*V*_DS_) provided further evidence that the photocurrent was generated with the exposure to UV illumination. The output characteristics showed typical n-type TFT behavior and demonstrated saturation behaviors as *V*_D_ became more positive, indicating that this device shows excellent gate voltage control of the drain current, as shown in Figure 4e. Moreover, to confirm the photoinduced current of the active channel layer film under UV light, the electrical performance was characterized with and without the UV light. It can be seen clearly that a significantly larger drain current was generated with an increase of more than 2 orders of magnitude when the UV light was turned on, as shown in Figure 4f. This increased off-current is attributed to the active channel layer of the photosensitive ZnO material, as well as to the photogenerated free electron carriers, which can be vigorously activated by UV light exposure. From the above results, UV light could generate a photocurrent immediately when no gate voltage was applied.

Figure 5a shows the diagram of the ZnO-based TFT photodetector under 5.06 μW of UV illumination (*V*_G_ = 0). Furthermore, this photodetector showed excellent gate electric field control under UV light exposure, with the photocurrent *I_photo_ = (I_light_ − I_dark_)* increase associated with the gate bias and drain bias increasing, as shown in Figure 5b. This efficient current amplification was associated with the negative charge carriers generally accumulated at the channel layer/gate dielectric interface by an induced gate-bias field. Without any bias or with only a small gate voltage bias applied, the UV light induced only a small photocurrent with the photocarriers that had high enough energy to overcome the band gap in the ZnO from the valence band migrating to the conduction band, as shown in Figure 5c. However, the photocurrent could be amplified by the electric field applied across the drain and source as the drift current. (*I*_drift_ = σE, where σ is the conductivity of the ZnO channel and E is the electric field, V_G_) [39]. Under the application of a positive gate electric voltage, an electron accumulation region was formed inside ZnO in the vicinity of the dielectric layer. The spatial difference of the electron density gave rise to a larger photocurrent generation, as illustrated in Figure 5d. Therefore, the photocurrent could be amplified by the excellent drain-source bias and gate bias voltage control.

### 3.4. Plasmon TFTs of ZnO-Based Visible Light Photodetector Performance Characterization

As shown in above results, the ZnO-based TFTs of electrical characteristics showed no response under visible light illumination, indicating that the visible light cannot induce a photocurrent and threshold voltage shift because the photo energy was smaller than the band gap of ZnO. In order to extend the photoresponse under visible light, gold nanoparticles (AuNPs) were linked at the interface between the ZnO semiconducting layer and dielectric layer by 3-mercaptopropyltrimeth-oxysilane (MPTMS) with a concentration of 126 pM, as shown in Figure 6a. Appendix A shows a scanning electron microscope (SEM) image of AuNPs distributed in the channel region of ZnO between the source and drain electrodes. As confirmed by the UV-visible (UV-vis) absorption spectrum shown in Appendix A, the gold nanoparticles absorption maximum peak was 520 nm which corresponded to diameter size was 13 nm, and determined the gold nanoparticles were working mechanism for plasmon resonance energy detection at green light range wavelength. In terms of optical properties, the AuNPs with 13 nm in diameter exhibit strong plasmonic absorption around a light wavelength of ~520 nm.

Figure 6b,c shown the photo-detecting under visible light illumination (520 nm wavelength) in the transfer curves of the ZnO-based TFTs electrical characteristics of typical *I*_DS_–*V*_GS_ and the square root of *I*_DS_–*V*_GS_ curves, and the “initial state” represents the device that was not exposed to light illumination. As shown in Figure 6b of typical *I*_DS_–*V*_GS_ curves, a slight increased drain current (I_D_) was clearly observed with exposure of a halogen lamp’s light. As shown in Figure 6d which amplified saturation current interval from Figure 5b between gate voltage 10~20 volt, a slight increment of saturation current was clearly observed when the AuNPs absorbed the 520 nm light wavelength. Additionally, the device showed a significant variations negative shift of threshold voltage under different light power density with the exposure of visible light. The threshold voltage shifts from darkness −8 V to −10.5 and −12 V under 15.23 and 28.31 μW, respectively, of halogen lamp light illumination without any accompanying field-effect mobility degradation. In addition, the electrical properties were recovered to the initial state after the light was turned off.

This visible light photodetecting is based on plasmonic energy detection capability which was incorporated with gold nanoparticles (Au NPs) on the active semiconducting layer of ZnO. This photodetecting working principle is that the Au NPs behaved plasmonically, the metal nanostructure was of 13 nm diameter, and AuNPs absorbed visible light of 520 nm wavelength that induced the surface plasmon resonance (SPR) which leading to the electrical threshold voltage shift to negative direction characteristics and hot electrons inject in the active channel layer from the AuNPs metal nanostructure under visible light illumination, as shown in Figure 7. Furthermore, the visible light reaches the AuNPs which penetrates the transparent ZnO layer induces the SPR, the SPR not only leading the threshold voltage shift but also these hot electrons contribute to slight amplification of the saturation drain current by modulating the ZnO channel conductivity. Therefore, this photodetecting with AuNPs was based on a metal-semiconductor junction structure enables plasmon energy detection by using the plasmonic effects is an efficient conversion of plasmon energy to an electrical signal. These results show that this metal-semiconductor structure enables direct conversion of the absorbed plasmon resonance energy from the AuNPs to the semiconducting channel layer leading to the ZnO-based TFTs’ photodetecting detects, and only the UV wavelength extended to visible light wavelength.

## 4. Conclusions

In summary, we have successfully demonstrated a new three-dimensional (3D) channel structure of a zinc-oxide (ZnO)-based thin-film transistor (TFT) fabricated by a direct solution-based patterning technique of ink-jet printing, utilizing the increase in the substrate temperature for rapid solution evaporation rate to narrow the channel width. Moreover, we obtained a significant improvement in electrical characteristics and stability by a simple and efficient method of oxygen plasma treatment. This device enabled highly efficient UV light detection with immediately excited electron-hole pairs generating the photocurrent in the photoactive ZnO channel layer. In addition, the photocurrent generation due to the UV photoinduced electrons could be coupled with a gate and source-drain bias for enhanced controlled amplification. Voltage bias modified the energy bands and facilitated the efficient migration of UV light-induced photocarriers, which assisted in overcoming the ZnO band gap to create free carriers, modifying the channel conductivity. Furthermore, the internal field in the photoactive channel layer also enabled the efficient collection of these photocarriers to increase the saturation current, and the electrical characteristics to recover to the initial state without trapping the electron carriers in the dielectric layer. Additionally, we also have successfully demonstrated that a metal-semiconductor junction structure that enables plasmon energy detection by using the plasmonic effects is an efficient conversion of plasmon energy to an electrical signal. Leading to the ZnO-based TFTs, only ultraviolet light detection extends to the visible light wavelength. This study provides a method for the creation of a 3-D channel structure by the ink-jet printing technique and comprehensively demonstrates how the variations in the channel morphology can affect the electrical properties for the fabrication of TFTs with the aim of manufacturing large-area electronic devices; such devices will be useful in many photo-absorption detection techniques for UV and visible light wavelength (520 nm), as well as for studies of excitons and charge transport in optoelectronics.

## Figures and Tables

**Figure 1 nanomaterials-10-00458-f001:**
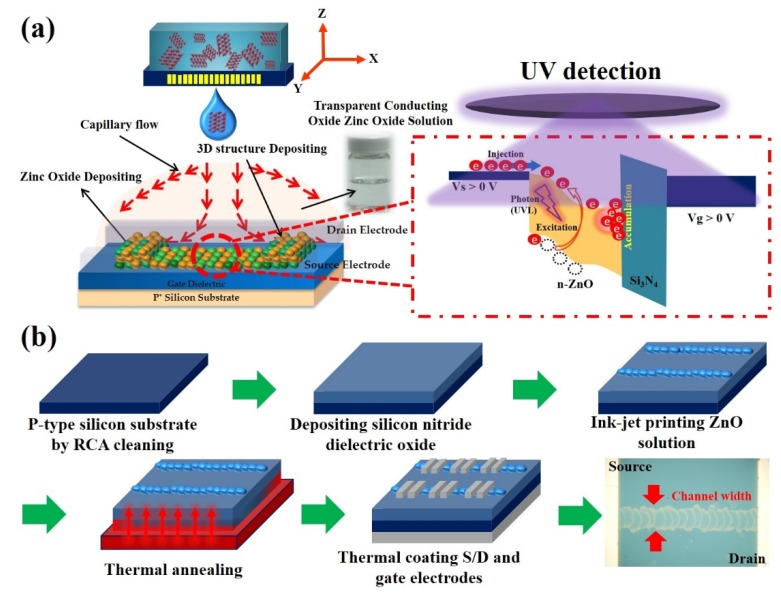
Schematic of ink-jet printing of ZnO thin-film transistors (TFTs) for a ultraviolet (UV) photodetector. (**a**) The ink-jet printing three-dimensional channel structure induced by the coffee ring effect and mechanism of the ultraviolet light-induced photocurrent in the photoactive of ZnO film. (**b**) The fabrication procedure of ZnO-based TFTs by ink-jet printing with low-temperature processing.

**Figure 2 nanomaterials-10-00458-f002:**
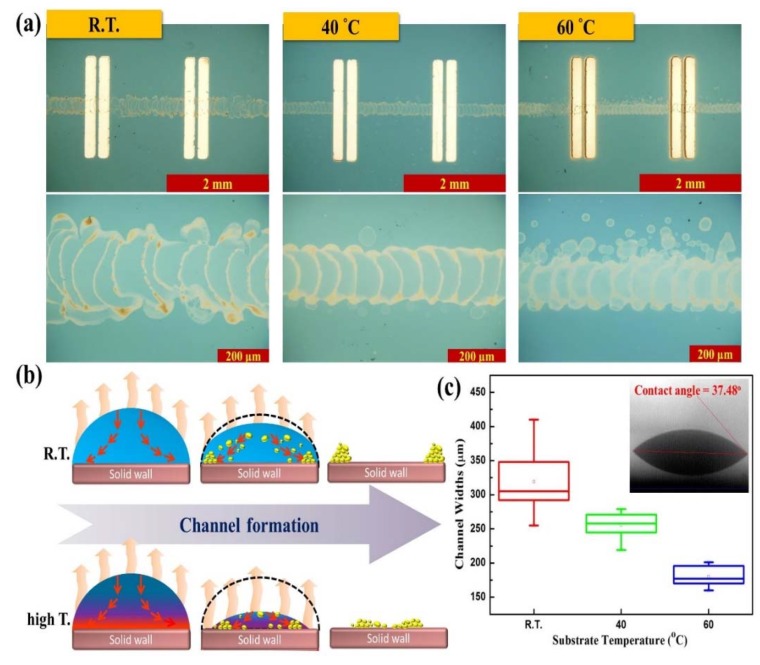
The 3-D semiconducting channel structure of ink-jet-printed ZnO-based TFTs. (**a**) Optical microscope (OM) images of the ZnO remaining on the substrate after the solution evaporation at different substrate temperature, optical magnification in (×4/0.13, top) and (×20/0.50, bottom), respectively. (**b**) Schematic diagrams of the evaporation phenomenon in a droplet at different temperatures: room temperature (top) and higher temperature (bottom). (**c**) The channel widths were normalized for 10 devices by an IJP droplet with the substrate temperature at room temperature, 40 °C and 60 °C, respectively. The inset shows the contact angle analysis of the dielectric layer, which was performed using a DI water droplet.

**Figure 3 nanomaterials-10-00458-f003:**
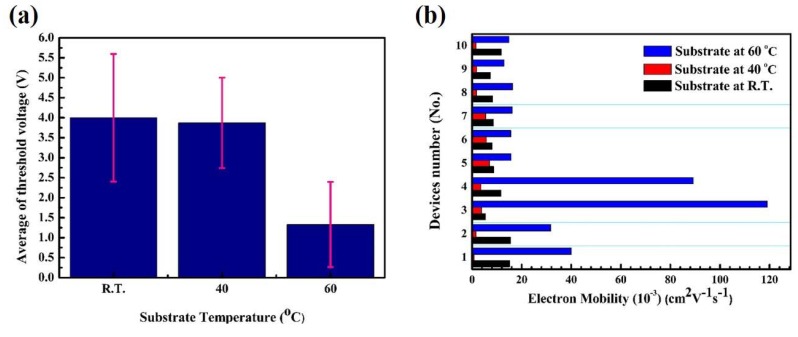
Electrical characterization of the ZnO-based TFTs through ink-jet printing in the atmosphere analyzer for measurements performed in air, where I_DS_ was measured while *V*_GS_ was scanned from −20 V up to 20 V at a constant *V*_DS_ of 5 V. (**a**) Comparison of the average values for the absolute normalized threshold voltage (*V*_th_) of the TFTs prepared at room temperature, 40 and 60 ºC; electrical characterization of the average values was achieved from 10 TFT devices. (**b**) Comparison of the distribution of saturation electrons mobility for 10 ink-jet printed TFTs. (**c**) Typical transfer characteristics of *I*_DS_–*V*_GS_. (**d**) X-ray photoelectron spectroscopy (XPS) spectra of the O1s region of ZnO films with/without oxygen plasma treatment.

**Figure 4 nanomaterials-10-00458-f004:**
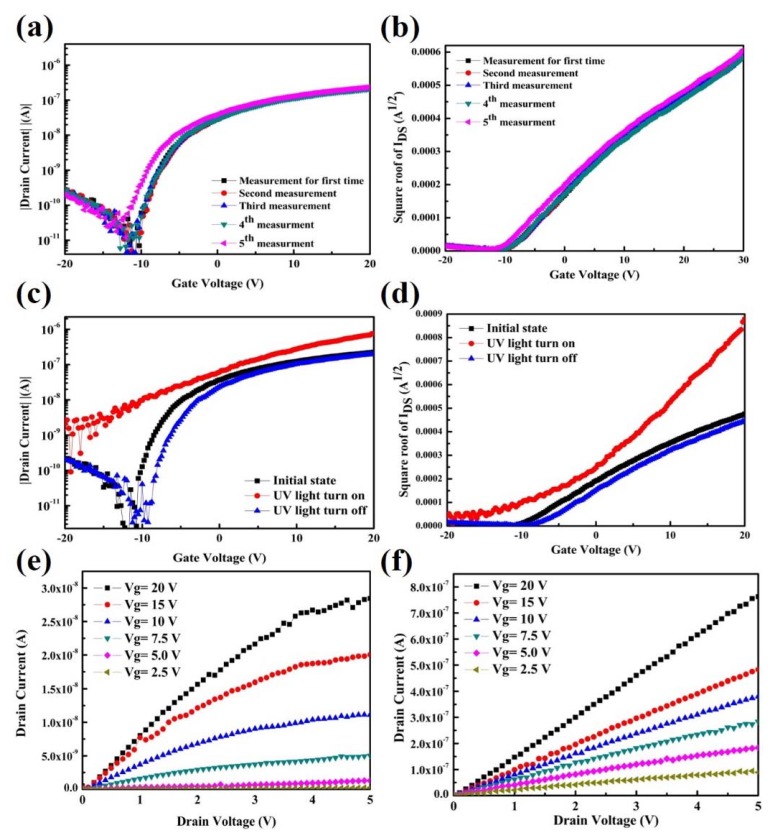
Electrical characteristics of ultraviolet light photodetector based on ZnO thin-film transistors. (**a**) Electrical performance endurance test performed 5 times by applying a drain current with the bottom gate voltage varying from negative 20 to positive 20 V in steps of 250 mV and the drain voltage fixed at 5 V and (**b**) square root of the drain current-gate voltage transfer characteristic curves. (**c**) The *I*_DS_–*V*_GS_ curves of the photodetector under darkness, under UV illumination and with the UV light turned off and (**d**) square root of the electrical characteristics. (**e**) Typical drain current as a function of the drain voltage with the bottom gate voltage varying from 2.5 to 20 V under darkness and (**f**) UV illumination.

**Figure 5 nanomaterials-10-00458-f005:**
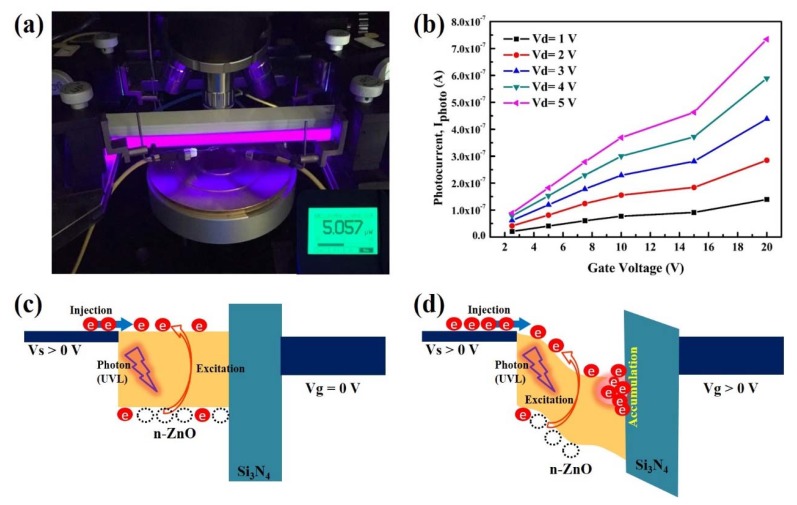
(**a**) Photograph of the ZnO-based photodetector on a semiconductor measuring the load under UV illumination. (**b**) Photocurrent increase associated with the gate bias and drain bias increase. The gate bias influenced the ZnO energy band bending of the photodetector and amplification mechanism. (**c**) Photocurrent induced by UV light without gate bias and (**d**) with gate bias voltage.

**Figure 6 nanomaterials-10-00458-f006:**
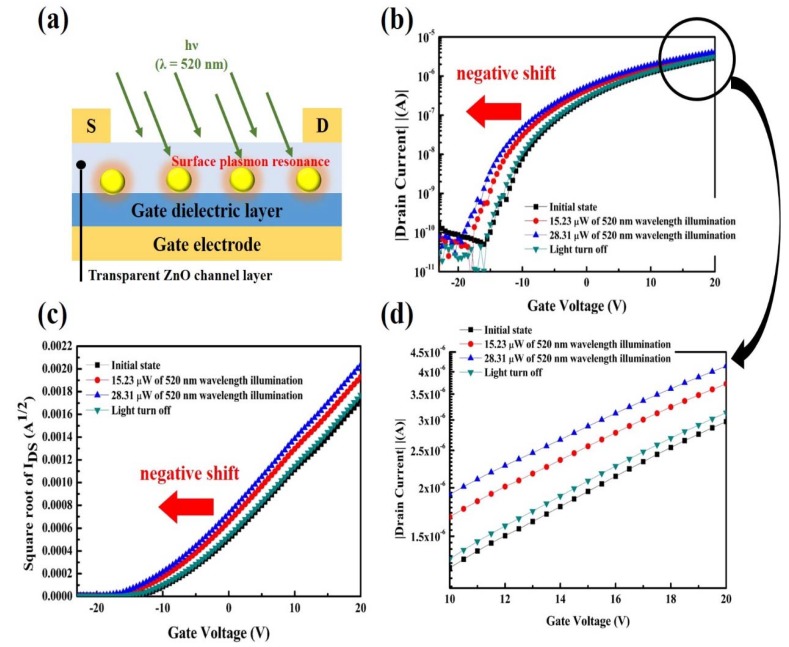
(**a**) Schematic illumination of the device, where Au NPs are inserted at the interface between transparent ZnO channel layer and gate dielectric. (**b**) The I_DS_–V_GS_ curves of the photodetector under darkness (initial state), under visible light of 520 nm wavelength illumination and with the light turned off and (**c**) square root of the electrical characteristics. (**d**) Amplified saturation current interval from Figure 6b between gate voltage 10~20 volts.

**Figure 7 nanomaterials-10-00458-f007:**
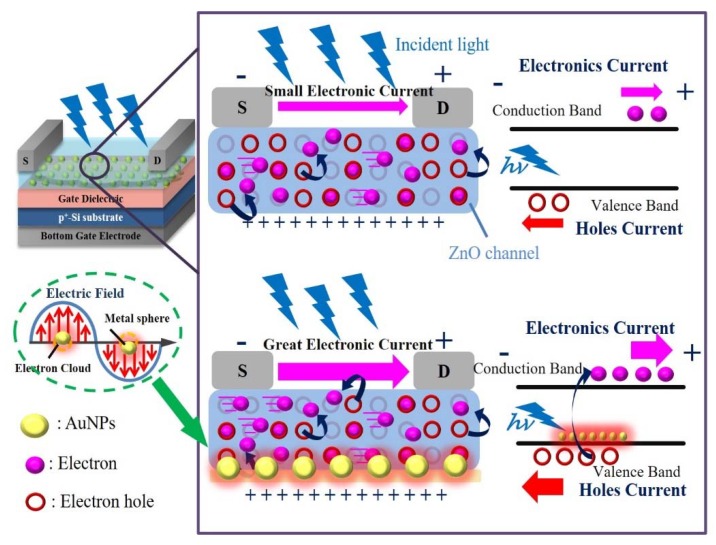
The ZnO-based TFTs incorporated with AuNPs that induced surface plasmon resonance when they absorbed the 520 nm wavelength of light illumination.

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
