# Peer review of "Ultraviolet Photodetecting and Plasmon-to-Electric Conversion of Controlled Inkjet-Printing Thin-Film Transistors"

_nanomaterials, 2020, doi:10.3390/nano10030458_

Round 1
Reviewer 1 Report
In this paper, a lot of very interesting results are demonstrated.
The research is undoubtedly original and promising. The results are abundant and precise. But the §3 ("Results and Discussion") and more particularly the subsection 3.3 ("ZnO-based TFT of UV and visible light photodetector performance characterization") should be more structured in order to facilitate the reading. A suggestion : could you re-organize this section in order to better highlight the main results ?
The scientific content of this article should be better emphasized.
Author Response
Thank you very much for your comment on our work. In order to make the manuscript easier to understand and facilitate the reading, we polished entire manuscript and highlighted the main results in this article, especially modified the section of abstract, introduction, results and discussion, and re-organized the structure in section 3. Please find the "Track Changes" in the revise manuscript. The English style has been polished by professional English editing service of AJE. We believe that this article after modification is more fluent scientific discourse.

Reviewer 2 Report
The study successfully describes a new ZnO based structure that is a thin film transistor fabricated by a patterning technique. As it is demonstrated, such a device is able to detect UV light and generate a current of electrons, that can be amplified.
As conclusion, the study provides a new method for the synthesis of this materials and the study of the electrical properties/applications, I recommend its publication.
I would recommend to revise the text, and specially the abstract and the introduction sections, since it is confusing and some parts are not clearly presented.
Author Response

(The authors gave the same response as above.)

Reviewer 3 Report
This paper reports the ZnO based TFT device fabricated by the inject process. It is demonstrated that TFT arrays can be made by the use of inject printing of ZnO precursor on SiN substrate. It is a significant achievement, especially the device performance reported, and many scientists will find this paper to be of great importance. There are a couple of minor comments to make for potential improvement.
1) editorial: there are occasional mis-spell and grammatical errors. Careful proofreading should be conducted. Also check the consistency of the statements. For example, the introduction says "three key" points of findings but in reality there are four.
2) threshold voltage: one of the interesting findings of this work is related to the threshold voltage that is high for room temperature and 40C evaporation. On the other hand, it shows a dramatically lowered voltage when ZnO precursor was dried at 60C (the variation among array is also significantly more). A certain variation in V within an array and lowering V with higher temperature is somewhat expected because the ZnO formation may not proceed in less uniform manner and also because the degree of microcrystalline phase formation may be greater (so resulting in significantly lowered V) Without specific mechanism explored or explained, it is difficult to see the reason behind such results while it is one of the most important and interesting findings. Exploration on the mechanism may be helpful for readers.
3) XPS: change in surface chemistry of ZnO by oxygen plasma treatment is expected. the question is how much. Fig.3-d shows the increase in O3. However, it is not easy to judge if O3 is a result of O1 conversion or O2 conversion. It may be best to quantitatively compare intensity of each specie and show the ratio in numbers.
Author Response
We are deeply thanks for your valuable comments. We have included a new figure S3, and please refer to the attached file for detailed revision.

Round 2
Reviewer 1 Report
Thank you for the improvement of the manuscript